# A Platform of Patient-Derived Microtumors Identifies Individual Treatment Responses and Therapeutic Vulnerabilities in Ovarian Cancer

**DOI:** 10.3390/cancers14122895

**Published:** 2022-06-12

**Authors:** Nicole Anderle, André Koch, Berthold Gierke, Anna-Lena Keller, Annette Staebler, Andreas Hartkopf, Sara Y. Brucker, Michael Pawlak, Katja Schenke-Layland, Christian Schmees

**Affiliations:** 1NMI Natural and Medical Sciences Institute, The University of Tuebingen, 72770 Reutlingen, Germany; anna-lena.keller@nmi.de (A.-L.K.); katja.schenke-layland@nmi.de (K.S.-L.); 2Department of Women’s Health, Eberhard Karls University Tuebingen, 72076 Tuebingen, Germany; andre.koch@med.uni-tuebingen.de (A.K.); andreas.hartkopf@med.uni-tuebingen.de (A.H.); sara.brucker@med.uni-tuebingen.de (S.Y.B.); 3NMI Technologie Transfer GmbH, Pharmaservices Protein Profiling, 72770 Reutlingen, Germany; berthold.gierke@nmi.de (B.G.); michael.pawlak@nmi.de (M.P.); 4Institute of Pathology and Neuropathology, Eberhard Karls University Tuebingen, 72076 Tuebingen, Germany; annette.staebler@med.uni-tuebingen.de; 5Department of Gynecology and Obstetrics, University Hospital of Ulm, 89081 Ulm, Germany; 6Cluster of Excellence iFIT (EXC2180) ”Image-Guided and Functionally Instructed Tumor Therapies”, Eberhard Karls University Tuebingen, 72076 Tuebingen, Germany; 7Department of Biomedical Engineering, Eberhard Karls University Tuebingen, 72076 Tuebingen, Germany; 8Department of Medicine/Cardiology, Cardiovascular Research Laboratories, David Geffen School of Medicine at UCLA, Los Angeles, CA 90095, USA

**Keywords:** patient-derived tumor model, ovarian cancer, anti-cancer drug sensitivity, RPPA protein profiling, cancer immunotherapy

## Abstract

**Simple Summary:**

For personalized oncology, it is crucial to develop appropriate patient-derived tumor models that allow individualized validation of the most effective cancer therapy. The objective of this study was to develop and characterize a new patient-derived ovarian cancer tumor model composed of patient-derived microtumors (PDM) and autologous tumor-infiltrating lymphocytes (TIL). In contrast to other preclinical tumor models, such as patient-derived organoids, PDM are generated within 24 h from fresh ovarian tumor samples. From immunohistochemical comparison with the original primary tumor, we conclude that the histopathological features of the original tumor are essentially preserved. Importantly, we successfully identified treatment-sensitive and treatment-resistant tumor models for standard platinum-based therapy by reverse-phase protein array (RPPA) analysis of PDM. Furthermore, we were able to evaluate the efficacy of cancer immunotherapy by co-culturing PDM and autologous TILs. PDM and TILs may therefore serve as a preclinical platform to identify individualized, tailored cancer treatments in the future.

**Abstract:**

In light of the frequent development of therapeutic resistance in cancer treatment, there is a strong need for personalized model systems representing patient tumor heterogeneity, while enabling parallel drug testing and identification of appropriate treatment responses in individual patients. Using ovarian cancer as a prime example of a heterogeneous tumor disease, we developed a 3D preclinical tumor model comprised of patient-derived microtumors (PDM) and autologous tumor-infiltrating lymphocytes (TILs) to identify individual treatment vulnerabilities and validate chemo-, immuno- and targeted therapy efficacies. Enzymatic digestion of primary ovarian cancer tissue and cultivation in defined serum-free media allowed rapid and efficient recovery of PDM, while preserving histopathological features of corresponding patient tumor tissue. Reverse-phase protein array (RPPA)-analyses of >110 total and phospho-proteins enabled the identification of patient-specific sensitivities to standard, platinum-based therapy and thereby the prediction of potential treatment-responders. Co-cultures of PDM and autologous TILs for individual efficacy testing of immune checkpoint inhibitor treatment demonstrated patient-specific enhancement of cytotoxic TIL activity by this therapeutic approach. Combining protein pathway analysis and drug efficacy testing of PDM enables drug mode-of-action analyses and therapeutic sensitivity prediction within a clinically relevant time frame after surgery. Follow-up studies in larger cohorts are currently under way to further evaluate the applicability of this platform to support clinical decision making.

## 1. Introduction

In the context of personalized medicine, patient-derived model systems are expected to play an important role in order to identify suitable and effective therapies for the individual patient as well as existing therapeutic resistances of the patient’s tumor. Especially for cancer types with dismal treatment success rates such as ovarian cancer (OvCa), these model systems will be valuable for future cancer therapy. OvCa is among the most lethal gynecological diseases in women, with >185,000 deaths worldwide in 2018 [1]. Late diagnosis and disease complexity characterized by strong molecular and genetic heterogeneity are causative for its poor survival rates and varying treatment response to first-line therapy. Substantial efforts have been made to resolve the complexity of OvCa, especially for high-grade serous carcinomas (HGSC) [2,3,4]. Despite the application of genomics and transcriptomics in elucidating disease determinants, the principles of responsiveness to therapy are still poorly understood [4]. The establishment of patient-derived tumor organoids (PDO) allowed addressing a number of these challenges for example by in-depth genetic and phenotypic tumor characterization and analysis of intra-tumoral heterogeneity in PDOs side-by-side with corresponding tumor tissue [5,6,7,8]. Even though recent studies have described the combination of PDO cultures with components of the tumor microenvironment including fibroblasts, endothelial cells and immune cells [9], PDOs do not fully reflect the original composition of primary tumor tissue in terms of extracellular matrix, tumor-associated fibroblasts, tumor-infiltrating lymphocytes (TILs), macrophages (TAMs), and tumor endothelial cells. Another challenge of current PDO models in terms of applicability for individualized drug response testing relates to the required establishment time of 1–3 months with a corresponding impact on the timeframe to obtain drug testing results [10]. Using OvCa as a prime model of a heterogeneous tumor disease, we introduce a three-dimensional (3D) preclinical ex vivo model composed of patient-derived microtumors (PDM) as well as autologous tumor-infiltrating lymphocytes (TILs) extracted from primary OvCa tissue specimen in a clinically relevant time-frame. Importantly, PDM recapitulate a 3D histo-architecture with retained cell–cell contacts and native intra-tumoral heterogeneity featuring the corresponding primary tumor microenvironment (including extracellular matrix proteins, stromal fibroblasts and immune cells). In combination with functional compound efficacy testing and multiplexed TILs phenotyping, we demonstrate the correlation of individual OvCa PDM responses to chemotherapeutic as well as immunotherapeutic treatment approaches using OvCa PDM alone and in co-culture with autologous TILs, respectively. We apply reverse-phase protein array (RPPA) analysis to map protein-signaling pathways of PDM and to measure on- and off-target drug effects in compound treated PDM. Albeit based on a small patient cohort the available clinical follow-up data suggests a correlation of obtained treatment responses in OvCa PDM models and corresponding patients indicating prolonged metastasis-free survival of identified carboplatin responders as compared to non-responders.

Based on the data presented here, we envision that our preclinical assay system combining PDM, autologous TILs and protein signaling pathway profiling could aid clinical decision making in the future and assist in the pre-selection of a personalized clinical treatment strategy for OvCa.

## 2. Materials and Methods

### 2.1. Human Specimens

Ovarian tumor samples were obtained from nineteen patients diagnosed with ovarian cancer undergoing surgery at the Center for Women’s Health, University Hospital Tuebingen. Written informed consent was obtained from all participants. The tumors were classified according to International Federation of Gynecology and Obstetrics (FIGO) grading system. Tumor samples were delivered on the day of operation. The research project was approved by the ethics committee (IRB#275/2017BO2 and IRB#788/2018BO2).

### 2.2. Isolation and Cultivation of Patient-Derived Microtumors and Tumor-Infiltrating Lymphocytes

The procedure was adapted from Kondo et al. (2011) [11] and modified as follows. Tumor specimens were washed in HBSS (Gibco, Thermo Fisher Scientific, Waltham, MA, USA), minced with forceps, and digested with LiberaseTM DH [12] for 2 h at 37 °C. Digested tissue was centrifuged (300× *g*, 5 min), washed with HBSS and filtered through a stainless 500 μm steel mesh (VWR). The flow-through was again filtered through a 40 μm cell strainer (Corning, Corning, NY, USA). The filtrate containing the TIL fraction was resuspended in Advanced RPMI 1640 (Gibco) supplemented with 2 mM Glutamine (Gibco), 1% MEM Vitamins (Gibco), 5% human serum (Sigma-Aldrich, St. Louis, MO, USA) and 100 μg/mL primocin (Invivogen, San Diego, CA, USA). IL-2 (100 U/mL), IL-7 (10 U/mL) and IL-15 (23.8 U/mL) (Peprotech, East Windsor, NJ, USA) were freshly added to culture media. For expansion, CD3/CD28 dynabeads were added (Milteny Biotech, Auburn, CA, USA). PDM, held back by cell strainer, were washed in HBSS and cultured in suspension in StemPro^®^ hESC SFM (Gibco) supplemented with 8 ng/mL FGF-basic (Gibco), 0.1 mM β-mercaptoethanol (Gibco), 1.8% BSA (Gibco) and 100 μg/mL primocin (Invivogen) within cell-repellent culture dish (60 × 15 mm) (Corning).

### 2.3. RPPA and Protein Data Analysis

Detailed methods of sample preparation and RPPA processing are provided in Appendix A. RPPA protein analysis and protein data processing was applied as reported before [13,14,15,16]. From the arrays, PDM sample signals were extracted as protein-normalized, background-corrected mean fluorescence intensity (NFI), as measured from two technical sample replicates. NFI signals, median-centered for each protein over all measured samples (including OvCa PDM and BC PDM samples) and log_2_ transformed, reflect a measure for relative protein abundance. Small NFI protein signals at around blank assay level (0.02 NFI) were as a limiting quality criterion excluded from further analysis; otherwise, all NFI signals were used for further protein data analysis. Protein heat maps were generated and cluster analysis (HCL) performed using the freely available MultiExperiment Viewer (MeV) software. For the comparison of protein profiles of treatment responders and non-responders (defined by functional compound testing), only proteins with a >20% difference between the means were used for analysis. On- and off-target pathway effects were evaluated from one biological and two technical replicate samples per model at three different treatment times (0.5, 4 and 72 h). Treated sample to respective DMSO vehicle control NFI ratios (TR) were calculated for each treatment condition and log_2_-transformed. A treatment-specific threshold of protein change (carboplatin: minimum 50% difference) was set. Only proteins showing treatment effects above the threshold were shown.

### 2.4. Efficacy of Compounds Validated in PDM Cultures

Efficacy of compounds was validated by applying the real-time CellTox™ Green Cytotoxicity assay (Promega). Assays were performed according to manufacturer’s protocol. PDM were cultured a maximum of 1–2 weeks in PDM culture medium prior testing. Per treatment, three to eight replicates were performed using *n* = 15 PDM per replicate in a total volume of 150 µL phenol-red free PDM culture medium. Cell death was measured as relative fluorescent unit (RFU) (485–500 nm Excitation/520–530 nm Emission), relative to the number of dead, permeable cells after 24 h, 48 h and 72 h with the Envision Multilabel Plate Reader 2102 and Tecan Spark Multimode Plate Reader. RFU values were normalized to DMSO control according to used drug solvent. Treatment effects were measured as fold change (FC) compared to control. Differences between treated PDM and untreated PDM were calculated as fold change values separately for each time point. Statistical significance was evaluated by two-way ANOVA multiple comparison test. Outliers were identified with the Iglewicz and Hoaglin’s robust test for multiple outliers applying a recommended Z-score of ≥3.5 [17].

### 2.5. FACS Analysis

To characterize lymphocyte populations within autologous TIL, cells were harvested (up to 1 × 10^6^ cells/staining depending on available number of cells), washed 2× with PBS (200 rpm, 5 min at 4 °C), resuspended in staining buffer (PBS plus 10% FBS) and plated in a 96-well V-bottom plate (100 μL/well) (Corning). To verify >90% cell viability, cells were counted with a Nucleocounter (Chemotec) before plating. For each panel staining, an unstained control and, if necessary, a FMO control were prepared. For extracellular staining, cells were incubated with antibodies (see Appendix A) for 30 min at 4 °C in the dark. For subsequent intracellular staining, cells were washed 2–3 times (200 rpm, 5 min at 4 °C) in eBioscience™ Permeabilization buffer (250 µL/well) (Invitrogen) and resuspended in eBioscience™ Fixation/Permeabilization solution (Invitrogen) for 20 min at 4 °C. After 2–3 washing steps (200 rpm, 5 min at 4 °C), cells were incubated with antibodies (30 min, 4 °C in dark) (see Appendix A). After the staining process, cells were washed 2–3 times and analyzed with a BD FACS Melody machine (BD Biosciences, Franklin Lakes, NJ, USA).

### 2.6. Co-Culture of PDM and Autologous TILs

To measure if the expanded, autologous TILs are able to kill corresponding PDM, we performed endpoint killing assays in a 96-well format with an image-based analysis using Imaris 8.0 software. First, PDM were pretreated with IFNγ (200 ng/mL) for 24 h to stimulate antigen presentation. In parallel, 96-well plates were coated with 5 g/mL of anti-CD28 antibody (Biolegend) o/n at 4 °C to provide a co-stimulatory signal during co-culture. On the next day, coated plates were washed 3× with PBS. PDM were washed in HBSS, centrifuged and resuspended in co-culture assay media consisting of RPMI 1640 phenol red free (GIBCO) supplemented with 2 mM Glutamin (Gibco), 5% human serum (Sigma-Aldrich, St. Louis, MO, USA), 1× MEM Vitamins (Gibco) and 100 µg/mL Primocin (Invivogen). Prior to assembling the co-culture, TILs were labeled with CellTracker™ Deep Red Dye (Thermo Fisher Scientific, Waltham, MA, USA) to differentiate between PDM and TILs. Labeled TILs were then co-cultured with PDM and in the presence of selected checkpoint immune inhibitors (CPIs: Pembrolizumab, Atezolizumab, Ipilimumab; Selleck Chemicals GmbH) or control anti-IgG4 antibody with an E:T ratio of 4:1. Thereby we counted 200 cells per single PDM. Per condition, we prepared triplicates each with 15 PDM and 12,000 TILs per well. After 92 h, cells were incubated with a staining solution consisting of live cell stain Calcein-AM (Thermo Fisher Scientific) and Sytox™ Orange dead cell stain (Thermo Fisher Scientific). After 1 h, Z-stacks of *n* = 3 PDM per well were imaged using a spinning disk microscope (ZEISS CellObserver Z1). Only viable PDM were positively stained by Calcein-AM, while all dead cells were stained by Sytox™ Orange. TILs were filtered by CellTracker™ Deep Red signal. Using the Imaris 8.0 software, we applied three masks, one for dead cells, one for dead TILs and one for live PDMs. For each mask, the total sum of all fluorescent intensities (FI) was calculated and the following ratio determined:(1)$$\%\;ratio\;dead\;vs.\;viable\;PDM\;\left[FI\right]=\frac{total\;dead\;\left[FI\right]-dead\;TIL\;\left[FI\right]}{viable\;PDM\;\left[FI\right]}$$

### 2.7. Statistical Analysis

Statistical analysis was performed using GraphPad Prism. For Boxplot data, whiskers represent quartiles with minimum and maximum values and the median. Datasets with no clear normal distribution were analyzed with unpaired, two-tailed Mann–Whitney–*U*-test, otherwise as indicated. Correlation data were evaluated by Spearman’s rank correlation. For all analyses, *p* values < 0.05 were considered statistically significant. Recommended post hoc tests were applied for multiple comparisons.

## 3. Results

### 3.1. Isolation of Patient-Derived Microtumors with High Viability from Primary OvCa Tissue Specimen by Limited Enzymatic Digestion

Residual fresh tumor tissue samples were collected from *n* = 16 OvCa patients undergoing primary tumor debulking surgery. The PDM and TIL isolation procedure (further developed from Kondo et al. 2011) [11] was performed on freshly excised tumor tissue specimen (Figure 1).

Available anonymized clinico-pathological characteristics including International Federation of Gynecology and Obstetrics (FIGO) staging and pathological TNM-classification of respective individuals are summarized in Table 1. Overall, 2/19 patients (OvCa #4 and OvCa #18) received neoadjuvant treatment with carboplatin/paclitaxel chemotherapy. The majority of included samples (*n* = 15) were derived from the most common type of OvCa, i.e., epithelial OvCa, with a majority of high-grade serous carcinomas (HGSC). One sample was classified as sex cord–stromal ovarian carcinoma that is either non-malignant or at a low stage.

Isolation of PDM was successful in 87.5% (14/16) of the tumor samples (Figure 1A) with varying amounts of available PDM for downstream analyses such as live–dead staining, immunohistochemical characterization, protein signaling pathway analyses and efficacy drug testing of standard-of-care therapy as well as immunotherapy. PDMs were cultured in suspension in the absence of serum for a maximum of three weeks. No correlation was observed between successful isolation of PDM and available clinical patient data such as age, lymph node spread, distant cancer spread, perineural invasion or FIGO stage (Appendix A). PDM viability was assessed by parallel staining with Calcein-AM and SYTOX™ Orange (Figure 1B). The 2D projections of 3D images displayed highly viable PDM with few dead cells. Dead PDM cells (according to nuclear SYTOX™ Orange staining) detached from PDM and thus observed mostly as single cells floating in the culture media. The quantification of the viable cell volume and dead cell volume in 3D projections of four exemplary OvCa PDM models are shown in Figure 1C. In each analyzed model, ≤7% of the total PDM cell mass represented dead cells confirming robust PDM viability.

### 3.2. OvCa PDM Sections Display Histopathological Characteristics Comparable to the Corresponding Primary Tumor Tissue (PTT)

We next performed Hematoxylin and Eosin staining (H&E) of FFPE- and cryo-sections, respectively, derived from OvCa PDM and corresponding primary tumor tissue sections (PTT) for histopathological comparison. Professional assessment of PDM by a certified pathologist, confirmed the presence of typical, distinct histopathological characteristics of OvCa in respective PDM (Figure 2 and Appendix A). HGSC derived PDM reflected architectural patterns such as papillary growth, irregular branching, cystic and glandular structures (Figure 2 OvCa #17–23; Appendix A, OvCa #24, 26) comparable to the corresponding PTT specimen. Pleomorphic nuclei/cells, high nucleus:cytoplasm ratio as well as hyperchromasia were similar in PDM and corresponding PTT sections reflecting the high-grade of analyzed HGSC tumors. These tumor features were not detected within OvCa PDM #8 (Appendix A), which originated from low-grade mucosal OvCa known for slow tumor growth. Instead, OvCa #8 PDM displayed a unicellular epithelium and mostly stromal remains. In summary, histopathological analyses of PDM confirmed structural and cellular similarities to the corresponding primary tumor specimen and the conservation of typical histological features of ovarian carcinomas.

### 3.3. Immunohistochemical Staining of PDM Identifies Expression of Histopathological OvCa Markers and Patterns of Extracellular Matrix and Tumour Microenvironment Components Comparable to Corresponding Primary Tumour Tissue Sections

For further characterization of histological similarities and differences between OvCa PDM and corresponding PTT, the expression of histotype specific markers together with tumor cell-, extracellular matrix- and immune cell-markers were assessed by immunohistochemistry. The degree of marker-specific staining patterns in obtained images of PDM and PTT sections was quantified by image-based analysis (Figure 2 and Appendix A). In the clinics, immunohistochemical staining of p53 and Wilms Tumor 1 (WT1) is applied for differential diagnosis of HGSCs [18]. These two markers are the only ones examined in routine histopathology. In-depth histopathological assessment by a certified pathologist revealed that the HGSC phenotype of the original tumor persists in the corresponding PDM (see above). In line with this, expression of WT1/p53 in PDM corresponded well with either low-to-moderate (OvCa #17, #18 and #25) or strong expression (OvCa #23, #24 and #26) in respective PTT sections (Figure 2 and Appendix A). Except for OvCa #24, where PDM showed significantly pronounced p53 staining as compared to corresponding PTT, image-based analysis did not show significant differences between PDM and PTT for WT1.

Mesothelin (MSLN) and CA125 (MUC16) were investigated as additional OvCa markers.

Mesothelin, known to be over-expressed on the cell surface in OvCa [19,20,21], was found to be differentially expressed in four out of seven analyzed PDM models as compared to corresponding PTT sections, with OvCa PDM #17 and #18 displaying higher and OvCa PDM #23 and #26 showing lower MSLN expression as compared to respective PTT sections (Figure 2B and Appendix A).

For CA125, no significant difference in expression between PDM and PTT sections of the OvCa models studied here was observed. CA125 expression has previously been described as an immunohistochemical marker to confirm ovarian origin of the tumor [22]. As shown before [23,24], expression of CA125 in OvCa sections can vary within one type and between the different OvCa tumor types. Accordingly, PTT sections derived from non-HGSC displayed no CA125 expression (OvCa #8) in contrast to HGSC-derived tumor sections (OvCa #18 and #24). CA125 expression was low or not detectable within the other PDM models studied here. As the tumor microenvironment is known to play a major role in tumor progression and metastasis [25,26,27], we analyzed the presence of extracellular matrix (ECM) and stromal components in OvCa PDM and corresponding PTT. Sections were stained for FAPα (Fibroblast associated protein alpha), a marker of cancer-associated fibroblasts (CAFs). FAPα expression in tumor stroma is observed in 90% of human cancers of epithelial origin and has been described to induce tumor progression and chemoresistance [28]. FAPα expression was detectable in 7 out of 11 OvCa PDM models studied. The observed FAPα staining pattern did not significantly differ between PDM and corresponding PTT sections from 5 of 7 OvCa models analyzed in our study (Figure 2 and Appendix A). OvCa PDM #17 displayed a significantly higher expression of FAPα as compared to the corresponding PTT sections. In contrast, for OvCa PDM #25 a lower degree of FAPα expression compared to respective PTT sections was observed.

Expression of the ECM component Collagen I, known to promote invasiveness and tumor progression in epithelial OvCa [29], was also prominent within OvCa PDM. Except for OvCa #18, #23 and #25, where a lower expression of Collagen I was observed in PDM as compared to corresponding PTT, the observed Collagen I staining pattern in PDM did not significantly differ from respective PTT sections.

In addition, we observed a correlation of the expression of another ECM component (Hyaluronan Binding Protein 1 (C1QBP)) in PDM and corresponding PTT for the majority of analyzed samples except for OvCa #25, where PDM expressed significantly lower levels of this marker as compared to respective PTT sections (Figure 2 and Appendix A). C1QBP is known to interact with the major ECM component hyaluronan [30].

In summary, the analyzed stromal and ECM components were found to be expressed in the majority of generated OvCa PDM models. In most cases, the observed expression pattern of respective markers in PDM did not significantly differ from expression in corresponding PTT sections.

To further examine tumor microenvironment (TME) components of PDM, we studied the infiltration with tumor-associated macrophages (TAMs) via CD163 expression together with the expression of the inhibitory checkpoint receptor ligand PD-L1. IHC analyses rarely detected M2-like TAMs (CD163+) within PTT and PDM sections and if so, mostly in stromal tissue parts of PTT. While macrophages were abundant in OvCa #24 PTT, they were not detected in the corresponding PDM (Appendix A). In contrast, for OvCa #17, CD163+ TAMs were detected in both PDM and PTT sections (Figure 2). Immune checkpoint receptor ligands are known to be expressed on tumor and/or immune cells of the tumor microenvironment. Here, PD-L1 expression was mostly absent in OvCa PTT and corresponding PDM sections.

In conclusion, parallel immunohistochemical staining of OvCa PDM and corresponding PTT sections showed their comparability for the majority of samples with PDM regarding features of the original tumor including presence of ECM and TME components together with expression of p53 and WT1 as markers important for the histopathological assessment of OvCa. In comparison with corresponding PTT sections, pure stromal areas were mostly absent from stained PDM sections, which might explain abovementioned differences in marker expression observed between PDM and corresponding PTT with regard to immune cell infiltration and degree of expression of stromal components. Moreover, in PTT, expression of CA125 and MSLN appeared to be mostly restricted to tumor cells at the margin of the stroma, which are less detectable within PDM.

### 3.4. Protein Signaling Pathway Profiling of OvCa PDM by RPPA

After initial immunohistochemical characterization of the 3D OvCa PDM that confirmed the presence of TME components in PDM similar to corresponding PTT, we performed an in-depth examination of the heterogeneity and molecular composition of different OvCa PDM models by generating signaling pathway protein profiles using RPPA. Protein abundances of 116 different proteins (including total and post-translationally modified forms) were measured in OvCa PDM samples each with a sample size of *n* = 100–150 per individual PDM (Figure 3A). One further PDM sample derived from human BC (breast cancer) was included to scale up the protein sample data and for comparison as both cancer types are known to share molecular and microenvironmental similarities (26, 30). Obtained protein-normalized, background-corrected mean fluorescence intensity (NFI) signals were median-centered to all samples (*n* = 8) and log_2_ transformed. Protein profiles of PDM samples covered signaling pathways such as for cell cycle, DNA damage response, apoptosis, chromatin regulation, MAPK/RTK, PI3K/AKT with mTOR, Wnt and NFκB, as well as OvCa tumor/stem cell markers. By hierarchical clustering (HCL), PDM samples were grouped according to their similarities in relative protein signal intensity (Figure 3A). Data analysis revealed three clusters: (1) OvCa #21 (OvCa granulosa cell tumor) and #23 (HGSC), with the most distinct protein profiles as compared to the other PDM analyzed; (2) OvCa #19 (HGSC) and the BC PDM shared more similarities than OvCa #19 with the other OvCa PDM models; (3). The remaining PDM samples resembled the third cluster with the most similar protein expression profiles containing exclusively HGSC models. Long distances of the sample dendrogram further underlines the proteomic heterogeneity of similar histopathological OvCa tumor types.

To compare protein abundances within different signaling pathways as well as of tumor/stem cell markers, proteins with impact on pathway activity were sorted according to their pathway affiliation (Figure 3B, Appendix A). Significant differences between PDM models were observed for the cell cycle pathway and the MAPK/RTK pathway. Highest cell cycle activity was found in OvCa #17 and #24 with almost 50% higher median NFI signals compared to OvCa #21 with the lowest median signals (median NFI = −0.33 log_2_) resembling a different histopathological tumor type compared to the other PDM models analyzed. MAPK/RTK pathway signaling was increased in OvCa #21 (median NFI = 0.38 log_2_), #23 (median NFI= 0.32 log_2_), #24 (median NFI = 0.31 log_2_) and #17 (median NFI = 0.30 log_2_). The BC PDM model was characterized by decreased median NFI signals of MAPK/RTK proteins (median NFI = −0.47 log_2_). Proteins related to PI3K/AKT pathway and of associated pathways were more abundant in OvCa #17 and #24. The mTOR pathway levels were elevated in OvCa #24 (median NFI = 0.54 log_2_) in other OvCa PDM this pathway showed comparable activity. Median NFI signals from apoptosis-related proteins were significantly different between OvCa #25 (median NFI = 0.75 log_2_) and BC PDM (median NFI= 1.41 log_2_). OvCa tumor/stem cell marker protein abundance was significantly upregulated in both OvCa #17 and #23 compared to BC PDM. Thus, RPPA protein profiling analysis demonstrated the heterogeneous activity of several signaling pathways within different OvCa PDM. Apoptosis-related proteins and OvCa tumor/stem cell marker proteins indicated the strongest differences between OvCa PDM models and the BC PDM model.

### 3.5. Heterogeneous Treatment Responses towards Chemo- and Targeted Therapy Assessed by Functional Compound Testing in OvCa PDM

Studies of targeted therapies in OvCa are often limited to clinical phase I and II or even cell-line-based preclinical studies [31,32,33], as treatment efficacies are heterogeneous and mostly not beneficial compared to standard chemotherapy. However, targeting specific signaling pathways could demonstrate a treatment alternative for individual OvCa patients either as first-line or recurrent cancer therapy. As we have discovered that protein abundances differed the most in the cell cycle and MAPK/RTK pathway in OvCa PDM (Figure 3A,B), we investigated efficacy of targeted inhibition of these pathways with the CDK4/6 inhibitor palbociclib, the MEK1/2 inhibitor selumetinib, as well as the Src-inhibitor saracatinib and compared these treatments to standard platinum-based chemotherapy (Figure 3C). PDM were treated with respective drugs, each at three different concentrations, chosen according to previously reported C_max_ concentrations [34]. Treatment efficacy in OvCa PDM—as measured by cytotoxicity—was heterogeneous among individual PDM models, with some specifically responding to carboplatin and others to targeted therapy. Carboplatin induced the most significant cytotoxic effects at the lowest dose (75 µM) at longest duration t = 72 h in OvCa #17 and #24 (Figure 3C). On the molecular level, RPPA protein profiling revealed significantly increased cell cycle activity in both models (Figure 3B), which might be associated with the stronger carboplatin response observed in OvCa PDM #17 and #24. Two additional PDM models were also carboplatin sensitive, but responded to treatment at higher dose (OvCa #23, #25). Accordingly, both had shown intermediate cell cycle activity in protein profiling analyses (Figure 3A,B). Selumetinib induced significant cell death in OvCa #17, #19, #21 and #23 at a final concentration of 100–150 nM (Figure 3C). The strongest effect was observed for OvCa #21, which displayed comparatively high MAPK/RTK pathway activity (Figure 3B). Palbociclib, an inhibitor of G1-cell cycle progression, caused significant cytotoxicity in OvCa #26, which had shown moderate cell cycle activity compared to the other models in RPPA protein analysis (Figure 3B). PDM models with significantly higher cell cycle activity as measured by RPPA (OvCa #17, #24), did not respond to palbociclib treatment. Inhibition of the Src-pathway by saracatinib caused significant and dose-dependent killing effects in OvCa #26. Saracatinib triggered rapid PDM death already after 24–48 h of treatment. In conclusion, functional compound testing further confirmed the molecular heterogeneity of studied OvCa PDM models identified by protein profiling. Interestingly, PDM models showing resistance to standard chemotherapy with carboplatin were instead sensitive towards targeted therapeutic approaches.

### 3.6. Correlation of Carboplatin Treatment Response and Activation State of Protein Signaling Pathways

With a focus on platinum-based standard-of-care chemotherapy, we next related the analyzed protein signaling pathways of untreated OvCa PDM to observed treatment responses. Therefore, protein NFI signals of PDM were grouped into responder and non-responder profiles according to significant carboplatin treatment effects from previously shown functional compound testing (Figure 3C). Mean protein signals (NFI) with >20% difference between responder and non-responder were plotted as a heat map, and significant differences between pathway signaling were analyzed. Further, we examined the on- and off-target pathway effects within different OvCa PDM models by RPPA to assess drug mode-of-action. For this aim, OvCa PDM were treated at one compound concentration and compared to vehicle (DMSO) control. Treatment-to-control signal ratios (TR) were determined from protein NFI signals of treated PDM samples and DMSO vehicle controls at three different time points for each treatment: immediate (30 min), early (4 h) and late (72 h). This enabled the exploration of fast and late treatment response based on changes of protein abundances within a given time frame.

#### 3.6.1. Carboplatin Treatment Sensitivity of OvCa PDM Correlates with High Protein Abundance of G2-M Cell Cycle Proteins

HCL clustering of PDM protein NFI signals led to five clusters that distinguish carboplatin sensitive and resistant PDM models (Figure 4A). To analyze significant differences related to activation or inactivation of signal transduction pathways, proteins from the HCL clustering were sorted according to their pathway affiliation and according to upregulation or downregulation in responder PDM models. Carboplatin-responder PDM models showed significantly increased cell cycle activity (*p* < 0.001; Figure 4B) with upregulated protein abundance observed for Aurora A kinase (mean NFI = 0.74 log_2_), CDK2 (mean NFI = 0.8 log_2_), Cyclin B1 (mean NFI = 0.84 log_2_), PCNA (mean NFI = 0.84 log_2_), and acetylated Tubulin (mean NFI = 0.1 log_2_) (Appendix A), which are mostly related to “mitosis” (35, 36). Phospho-Aurora A/B/C (Spearman’s r = 0.8827, *p* = 0.044), Cyclin B1 (Spearman’s r = 0.971, *p* = 0.011) and PCNA (Spearman’s r = 0.8827, *p* = 0.044) significantly correlated with carboplatin treatment sensitivity (Appendix A), which was graded according to recorded significance levels from “0–3” (“0”: *p* > 0.05; “1”: *p* < 0.05; “2”: *p* < 0.01; “3”: *p* < 0.001; Figure 3C). At the same time, carboplatin non-responder PDM models showed higher abundance of CDK1 (mean NFI = 0.38 log_2_), phospho-CDK2 (mean NFI = 0.77 log_2_) and phospho-CDK4 (mean NFI = 0.37 log_2_) (Appendix A), which are more related to the G0/G1 cell cycle phase. In addition, the apoptosis/DNA damage response pathway was significantly upregulated in carboplatin-responder compared to non-responder PDM models (*p* = 0.021; Figure 4B), especially with high abundance of cleaved caspase-8 and cleaved PARP (Appendix A). Additional significant differences between carboplatin-responder and non-responder OvCa PDM were detected within the RTK and the PI3K/AKT/NFκB signaling pathways (*p* < 0.001; Figure 4B). These pathways were downregulated in the carboplatin non-responder group. Higher EMT/tumor/CSC marker abundance was significantly associated with the carboplatin responder group (Figure 4B) including protein markers Mesothelin, Nanog, STAT1, and E-Cadherin (Appendix A). In contrast, there were few proteins found, which were down-regulated in the carboplatin responder group. Collectively, this panel of down-regulated proteins differed significantly compared to the carboplatin non-responder group (Figure 4B). It contained early cell cycle markers, e.g., Aurora A and Cyclin B1, the mTOR pathway effector phospho-S6RP, PDGFR and SNAI1. We further assessed metastasis-free-survival (MFS) between the described carboplatin-responder (OvCa #17, #23–25) and non-responder (OvCa #19, #26) PDM models (Appendix A). Metastasis-free-survival (MFS) analyses of available clinical follow-up patient data revealed prolonged median MFS of 16.2 months in carboplatin responder vs. versus 9.2 months in carboplatin non-responder models.

In summary, the activation state of different signaling pathways composed of proteins with >20% difference in abundance, allowed us to significantly distinguish carboplatin-responder from non-responder OvCa PDM models. Importantly, these protein signaling response profiles were well in line with results from functional compound efficacy testing assays using those OvCa PDM models.

#### 3.6.2. Carboplatin Treatment Is Associated with Early Induction of Stress Response and Late Apoptosis

Next, we sought to investigate the carboplatin drug mode-of-action within OvCa PDM. Therefore, the carboplatin-responding OvCa PDM #24 was treated with carboplatin at a concentration of 75 µM, which had significantly induced PDM cytotoxicity in this model (see Figure 3C). Protein NFI signals were measured at three different time points and normalized to vehicle control. Proteins revealing >50% difference in TR signals (Appendix A) were selected to focus on the strongest changes in abundance. Cell cycle progression proteins (phospho-CDK2, CDK1) and phospho-Histone H3 (Ser10), affecting chromatin condensation during cell division, were downregulated quickly within 30 min (Figure 4C). After 4 h of treatment, TR signals of phospho-Aurora A/B/C protein and Histone H3 was strongly increased (Appendix A). Longer incubation with carboplatin (72 h) resulted in strong downregulation of these proteins (Figure 4C). Diminished abundance of cell cycle proteins after 72 h of carboplatin treatment differed significantly from vehicle control (*p* < 0.001) and from early treatment (4 h; *p* < 0.001). While levels of cell cycle-related proteins decreased over time, apoptotic markers such as cleaved caspases as well as acetylated p53 were elevated after 72 h (Appendix A). Induction of apoptosis-related proteins was already observed after 4 h of treatment (Figure 4C) with increasing abundances of cleaved caspase-7 and acetylated p53 (Appendix A). Focusing on down-stream PI3K/AKT/mTOR/Wnt pathway regulation, the abundances of mTOR effector proteins (S6RP, S6RP-phospho) were quickly upregulated after immediate (0.5 h) carboplatin treatment (Appendix A), which is in line with previous reports about transcriptional regulation of stress response by the mTOR pathway [35]. We also observed additional elevation of mTOR pathway-related proteins after 4 h of carboplatin treatment. Furthermore, overactive mTOR signaling might have resulted in increased p53 activation through upregulated acetylated p53 levels (Appendix A) as described before [35]. The PI3K/AKT/mTOR pathway was significantly upregulated within 4 h of carboplatin treatment compared to vehicle control (*p* = 0.021; Figure 4C). Similar to proteins related to cell cycle, this pathway was completely abrogated as compared to vehicle control after 72 h of treatment (*p* < 0.001; Figure 4C). Changes in protein abundance differed significantly during all three measured time points (0.5 h vs. 4 h: *p* = 0.003; 4 h vs. 72 h and 0.5 h vs.72 h: *p* < 0.001 Figure 4C). Pronounced, significant downregulation of MAPK/RTK pathway occurred after 72 h of treatment (*p* = 0.017; Figure 4C). The observed proteomic changes within MAPK/RTK-related proteins over time were significant (0.5 h vs. 4 h: *p* = 0.009; 4 h vs. 72 h: *p* < 0.001; Figure 4C). Thus, carboplatin treatment of OvCa #24 illustrated substantial and time-dependent changes in TR signals. Short treatment with carboplatin apparently triggered the induction of stress responses while longer treatment duration caused the induction of apoptosis.

### 3.7. Characterization of Tumor-Infiltrating Lymphocyte Populations from Primary OvCa Tissue Samples

Our established procedure of tissue processing and PDM isolation enabled us to obtain single-cell suspensions containing tumor-infiltrating lymphocytes (TILs) from respective OvCa tumor specimen. This allowed for expansion of these autologous TILs in the presence of low-dosed cytokines and antigenic stimulation in order to investigate immuno-phenotypes of respective patient samples. The immunogenicity of OvCa has been demonstrated in prior studies and is confirmed by the frequent infiltration of ovarian tumors with TILs [36,37,38]. As reported by Sato et al. (2005), different T cell populations diversely influence tumor immunosurveillance in OvCa. High intraepithelial CD8^+^/CD4^+^ T cell ratios in patients were associated with improved survival as CD4^+^ T cells executed immunosuppressive functions. To determine the composition of the isolated immune cell infiltrate within our sample cohort, we characterized the phenotype of autologous TIL populations by multi-color flow cytometry (Appendix A). Within isolated and expanded OvCa TIL populations from different specimen, we found that the proportion of CD4+ TILs was 57.8% and significantly more abundant than CD8^+^ TILs with 33.5% (*p* = 0.003 **; Figure 5A, Appendix A).

#### 3.7.1. Isolated CD8^+^ OvCa TILs Are Composed of Tumor-Specific CD39^+^, Stem-like CD39^−^PD1^+^ and Terminally Differentiated CD39^+^PD1^+^ Populations

Within the isolated CD8^+^ TIL populations, we identified different phenotypes according to expression of the co-inhibitory receptors PD-1 and CTLA-4, the tumor-antigen specificity marker CD39 and the activation marker CD137 (Figure 5A). To investigate the activation status of CD8^+^ TILs, cells were examined for co-expression of the co-stimulatory receptor CD137 (4-1BB). CD137 is upregulated in activated T cells and has been suggested to be a marker for antigen-activated T cells [39]. The mean percentage of CD8^+^ CD137^+^ TILs was 3.1% and varied between 0–10% (Appendix A), and >5% of the CD8^+^ cytotoxic T-cells (CTLs) from OvCa #1, #3 and #5 (Figure 5B) co-expressed CD137, indicating their pre-exposure to tumor antigens. Expression of co-inhibitory receptors PD-1 and CTLA-4 on CD8^+^ TILs did not differ significantly among analyzed TIL populations but tended to higher PD-1 expression levels (mean 6.9% vs. 3.4%; Appendix A). TILs from OvCa #3, #7 and #25 as well as #5, #13 and #26 were among those displaying an exhausted phenotype with >10% of CD8^+^PD-1^+^ or CD8^+^CTLA-4^+^ TILs (Figure 5B). Moreover, in recent reports CD39 expression in CD8^+^ TILs was described as a marker for tumor-antigen specific TILs that have undergone tumor-antigen-driven clonal expansion, exhibit resident memory T cell-like phenotypes and express a variety of co-stimulatory and co-inhibitory receptors [40,41,42]. Here, CD39^+^ CTLs (mean 40.5%; range 4.4–96.8%, Appendix A) were significantly more abundant than CD39^−^ CTLs (mean 9.5%; range 0–48.3%, Appendix A), so-called ‘bystander TILs’, known to recognize mostly viral antigens (43) (*p* < 0.001, Figure 5A). The amount of CD39^+^ TILs strongly correlated with the amount of CD8^+^ TILs (Spearman r = 0.88, Appendix A; *p* < 0.001, Appendix A) and conversely with the amount of CD4^+^ TILs (Spearman r = −0.80, Appendix A; *p* = 0.002, Appendix A). Thus, the abundance of CD4^+^ and CD8^+^ TILs appeared to significantly determine the amount of CD39^+^ CTLs. In addition, CD39 expression was largely limited to CD8^+^ TILs. As co-inhibitory receptors play a role in T cell exhaustion and are important targets for immune checkpoint-inhibition, we analyzed PD-1 and CTLA-4 expression on the tumor-specific CD39^+^ CTL population. PD1^+^CD39^+^ were more frequent than CTLA-4^+^ CD39^+^ (15.7% vs. 5.4% Figure 5A, Appendix A). The total amount of CD8^+^PD1^+^ TILs thereby correlated with the amount of CD8^+^CD39^+^PD1^+^ TILs (Spearman r = 0.79, Appendix A; *p* = 0.002, Appendix A) of a PDM model. Thus, CD39 expression was limited to tumor-antigen-stimulated and -exhausted TILs (e.g., OvCa #7, #17 and #25; Figure 5B). In contrast to ‘terminally differentiated cells’ [43], OvCa TILs with a ‘stem cell-like’ CD39^−^PD1^+^ phenotype were found in 7.3% of the CTLs (Appendix A). This population showed the highest proportional variability with a maximum of 50.5% cells vs. a minimum of 0% as compared to other CD8^+^ TIL populations (CV 208%). The frequency of CD8^+^CD39^+^ and stem cell-like CD8^+^CD39^−^PD1^+^ was negatively correlated (Spearman r = −0.63, Appendix A; *p* = 0.024, Appendix A). These results confirm the feasibility of extracting and expanding TIL populations from fresh OvCa tissue samples and identify heterogeneous, patient-specific immuno-phenotypes with potential relevance for immuno-oncological treatment approaches.

#### 3.7.2. Specific TIL Phenotypes Isolated from OvCa Tumor Specimen Correlate with Regional Lymph Node Metastasis

We further analyzed the correlation between specific TIL populations and clinical follow-up patient data. OvCa patients with regional lymph node metastasis (*n* = 1) tended to present with significantly more extensive CD8^+^ TIL infiltration in their tumors than those with no lymph node metastasis (*n* = 0) (*p* = 0.016) (Figure 5C). Moreover, the frequency of CD8^+^ TILs appeared to significantly correlate with that of CD8^+^CD39^+^ TILs in OvCa (Figure 5C). Despite a small sample size, our data implicate a significant correlation between lymph node spread (*n* = 1) and the presence of a CD8^+^CD39^+^ population (*p* = 0.008).

### 3.8. OvCa PDM Killing by Autologous TIL Populations Is Enhanced by Immune Checkpoint Inhibitor Treatment

To evaluate the functional, tumor cell killing capacity of autologous TILs on OvCa PDM and the corresponding treatment efficacy of established immune checkpoint inhibitors (CPI), we subjected co-cultures of TILs and PDM from OvCa #24 and #26 to image-based analysis of CPI-treatment response. A total of nine PDM were imaged per treatment (three PDM per well in triplicates) and a dead:live PDM ratio was calculated according to the sum of measured fluorescent intensities (FI) (Figure 5D–F). Addition of TILs to autologous PDM induced a basal killing effect in PDM in both models analyzed in the absence of CPI treatment (Figure 5E,F). As the addition of matched isotype controls showed no additional effect in both co-culture models, we excluded the possibility of unspecific killing effects of CPI antibodies. TIL killing effects in OvCa #24 co-cultures were observed in response to treatment with either the combination of anti-PD1 and anti-CTLA-4 (pembrolizumab + ipilimumab) or anti-PD-L1 and anti-CTLA-4 (atezolizumab + ipilimumab) (*p* = 0.039) compared to isotype control treatment (Figure 5E). Single agents induced no significant increase in PDM killing. In OvCa #26 CPI treatment almost doubled PDM killing (Figure 5F). In comparison, co-cultures treated with ipilimumab (*p* = 0.004) or atezolizumab (*p* < 0.001) showed significant PDM killing compared to untreated PDM. The killing effect of TILs was significantly amplified by atezolizumab treatment compared to co-culture controls (PDM + TIL: *p* = 0.021; PDM + TIL + IgG4: *p* = 0.018; Figure 5F), In line with this observation, respective OvCa PDM models showed weakly positive PD-L1 staining (Appendix A). Further, atezolizumab treatment significantly increased the TIL killing effect towards PDM as compared to pembrolizumab (*p* = 0.026). Autologous CD8 TILs from both tested OvCa PDM models were composed of high amounts of tumor-specific, non-terminally differentiated CD8^+^CD39^+^ TIL populations as compared to other OvCa TILs (Figure 5B and Appendix A). Moreover, these CD8 TILs were prominently positive for CTLA-4, which might explain the observed increase in PDM killing in response to ipilimumab (Anti-CTLA4) treatment (Figure 5B and Appendix A). Thus, the co-culture of autologous TILs and PDM offers the possibility to extent compound efficacy testing beyond chemotherapeutic compounds to immune oncological treatment approaches in a patient-specific setting.

## 4. Discussion

Recently, we could show the establishment of PDM from human glioblastoma tissue specimen containing important components of the tumor stroma (e.g., tumor-associated macrophages), and their application for the assessment of responses towards CSF1R- and PD1-targeting antibodies as well as the small molecule inhibitor Argyrin F [44,45]. In the present study, we have now further extended this approach to a patient-derived model system composed of PDM and autologous TILs extracted from a panel of primary OvCa tissue specimen and their in-depth characterization by immunohistochemistry, protein profiling, immune cell phenotyping and focused compound efficacy testing. Our results show an 87.5% success rate for isolation of PDM with robust viability and in suitable amounts for further, multi-parametric downstream analyses. In-depth histopathological assessment of PDM sections by a certified pathologist confirmed the conservation of typical histological features of respective OvCa types by this model system. Importantly, the complexity of the ovarian cancer TME with respect to the presence of cancer-associated fibroblasts and extracellular matrix components including collagen and hyaluronan-binding protein observed in primary OvCa tissue sections was conserved and did not differ significantly in the majority of PDM models generated in our study. The presence of these TME components has previously been correlated with tumor stage, prognosis, and progression and shown to substantially influence treatment responses [29,46,47]. Moreover, our data show that PDM and corresponding PTT express similar levels of markers important for histopathological assessment of ovarian cancer such as p53, WT1 and CA125. Interestingly, we could also identify immune cell infiltration within a subset of OvCa PDM, reflecting the immunogenicity of OvCa as previously reported [36,37,48]. We also identified differences in protein expression between PDM and PTT (e.g., MSLN, Collagen or FAPα). This could be explained at least in part by the low proportion of pure stromal areas within PDM as compared to PTT.

While OvCa patient-derived organoids (PDO) were often studied by genomic and transcriptomic sequencing [6,7,8], we were the first (to our knowledge) to investigate inter-tumoral heterogeneity and differential drug response mechanisms by RPPA-based protein profiling in a patient-derived 3D OvCa preclinical cell model. Here, analyses of a panel of >110 phospho- and total proteins allowed for the clustering of histologically similar OvCa PDM models, pathway activity profiling and investigation of on- and off target drug effects. Obtained RPPA protein profiles confirmed the heterogeneity of OvCa PDM observed via immunohistochemistry and previously reported for HGSC, the most common type of OvCa. Our work identified significant differences in the activity of cell cycle and MAPK/RTK pathways within analyzed OvCa PDM and enabled their distinction from a breast cancer derived PDM model by differential expression of OvCa tumor and stem cell markers as well as apoptosis-related proteins.

Seven OvCa PDM models were applied for individualized compound efficacy testing using a panel of clinically approved drugs at C_max_ drug concentrations previously reported in clinical trials. For analyzed OvCa PDM models, we observed patient-specific heterogeneity of response towards chemotherapy and targeted therapy. Correlation with RPPA protein profiling data allowed the allocation of individual PDM drug responses to specifically up- or down-regulated signaling pathway activities and, importantly, enabled the prediction of PDM models with high probability of response towards chemotherapy or targeted therapy. In accordance with the ability of cytostatic drugs to induce apoptosis especially in actively dividing cells [49], our work identified additional correlation between proteins relevant for S and G2/M cell cycle phase progression and carboplatin response. Specifically, our data implicate that elevated abundances of Aurora A, Cyclin B1 and PCNA proteins may allow for identification of carboplatin treatment response. Furthermore and in line with previous reports, we confirmed that decreased DNA damage repair and the ability to undergo apoptosis [50] is associated with carboplatin treatment sensitivity in OvCa. This was illustrated by increased levels of cleaved caspase-7 and cleaved PARP. Our results did not identify a correlation of carboplatin resistance and markers of cancer stem cells (CSCs) [51,52] or epithelial-to-mesenchymal transition (EMT) [53,54]. Instead, we found the cancer stem cell-related protein Nanog as well N-Cadherin strongly upregulated in carboplatin-responding PDM. These differing results might arise from the fact that above-mentioned previous studies were performed in adherent cell lines and not within a patient-derived 3D tumor model. Importantly, we identified protein signatures of OvCa PDM allowing for the identification and prediction of PDM models with high probability of response towards chemotherapy or targeted therapy. The correlation of our results with clinical data indicated a significant correlation of carboplatin treatment response with prolonged metastasis-free survival of respective patients. Given the small sample cohort analyzed here, these results need to be interpreted with caution but warrant further investigation.

We further assessed proteomic changes upon PDM treatment such as effects on protein abundance, directed on- and off-target pathway effects and drug mechanism-of-action within OvCa PDM. In a carboplatin sensitive PDM model, we observed a time-dependent decrease in cell cycle- and an increase in apoptosis-inducing protein abundance. In parallel, we found a fast stress response upon treatment as indicated by an activated mTOR pathway with high S6RP and active phospho-S6RP levels [35]. Overactive mTOR in combination with cell stress and the inability of cells to adapt to cellular stress might be responsible for p53 elevation [55,56] and driving cells into senescence or apoptosis [57,58].

Apart from testing the response of OvCa PDM to conventional chemotherapy, we sought to investigate the applicability of this model system for efficacy assessment of immuno-oncological treatment approaches. For this aim, we applied immunophenotyping of autologous TIL populations followed by their co-culture with respective PDM in the presence and absence of immunotherapeutic mono- and combination treatment schedules. Immunosurveillance of cancer strongly depends on the composition of tumor-infiltrated immune cells and the degree of tumor tissue infiltration and is known to influence treatment efficacies. As a result, the idea of an immunoscore, identifying a patient’s immunophenotype, emerged [59]. Our work uncovered several immunophenotypes within expanded TILs from OvCa patients by multicolor flow cytometry compared to previous immunohistochemistry-based analysis [60]. As described by Sato et al. (2005) [36] and Zhang et al. (2003) [37] high numbers of intraepithelial CD8^+^ TILs are associated with better prognosis in OvCa. We found that OvCa TILs were largely composed of CD4^+^ rather than CD8^+^ TILs. In this regard, OvCa models with high amounts of suppressive CD4^+^ TILs and low numbers of CD8^+^ TILs are suggested to have worse prognosis [61]. In line with previous reports [62], we identified expression of CD39 in OvCa TIL populations, a marker that distinguishes between tumor-specific CTLs (CD39^+^) and bystander TILs (CD39^−^) [40,41]. Interestingly, we found that CD8^+^ TIL amounts correlated with that of CD8^+^ CD39^+^ TILs, and could confirm that these tumor-specific T cells constitute an exhausted, memory T cell-like phenotype, as CD39 expression was limited to CD8^+^PD-1^+^ TILs. Importantly, our results further demonstrated that co-cultures of PDM and autologous TILs could be applied to assess the treatment effect of CPIs in a preclinical and patient-specific setting. Such PDM-TIL co-culture systems could potentially be used to identify OvCa patients, who would most likely benefit from immunotherapies. In the limited OvCa tumor tissue cohort investigated here, OvCa tumors with regional lymph node metastasis contained higher numbers of CD8^+^ and CD8^+^CD39^+^ TILs. The co-culture models tested in our study for response towards CPI treatment were derived from lymph-node spreading primary tumors, which might suggest that immunogenicity of OvCa increases upon metastasis.

Our work illustrates several advantages of PDM over patient-derived cancer organoid (PDO) models. First, once the tissue sample is processed, PDM can be isolated within 2 days and used for various types of analyses including those we used here. In contrast, a period of weeks to months is usually required to establish a PDO line. Another advantage of PDM is their cellular composition and complexity with the presence of components of the ECM as well as the TME (including. Collagen, C1QBP, tumor-associated fibroblasts), which is more similar to the patient tumor than PDO, which are lacking these components [63]. Moreover, PDM are cultured in suspension, whereas PDO are usually cultured in ECM matrix from mouse tumors (Matrigel). Influences of animal origin and batch-dependent differences in the composition of matrigel on comparability with human tumor tissue need to be considered for PDO [64,65]. In contrast, PDM are cultured in defined medium without the addition of animal components.

Limitations of our PDM model are currently the restricted number of PDM available from digestion of individual tumor tissue samples. From experience with different tumor types, an average of several hundred to several thousand microtumors can be isolated from fresh tissue samples. This number depends on the amount of tissue available for PDM isolation as well as tissue composition (including degree of fibrosis and necrosis). PDMs are therefore presently not suitable for high-throughput drug screening approaches, but for focused drug testing in late preclinical and translational drug development as well as in the context of precision oncology. For our study, only a limited amount of corresponding primary tumor tissue was available for comparative analyses with isolated PDM. Comparative, RPPA-based analyses between PDM and PTT were not feasible here due to this limitation. Furthermore, the limitation of our present study with regard to sample size should be noted.

## 5. Conclusions

In conclusion, patient-derived microtumors isolated from OvCa tumor specimen represent a novel ex vivo tumor model for OvCa displaying histopathological similarities to corresponding primary patient tumors and revealing intertumoral heterogeneity as evidenced by immunohistochemical and protein profiling analyses. The combination of functional drug testing with analyses of protein signaling pathways and drug-mode of action enabled the identification of PDM models susceptible to platinum-based treatment and allowed for the prediction of individual therapeutic sensitivity. Parallel isolation and culturing of autologous TILs further allowed for the characterization of patient-individual immune-phenotypes as well as the assessment of responses towards immunotherapy in PDM-TIL co-cultures. While the rapid PDM/TIL extraction procedure and quick availability of resulting datasets within 3–4 weeks is in good accordance with timelines of clinical decision making, we plan to confirm our findings in future studies with larger sample cohorts. 

## Data Availability

All data needed to evaluate the conclusions of the paper are included in this published article and its Appendix A. Material and further data are available upon request after signature of an MTA from the corresponding authors.

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
