# Peer review of "A Platform of Patient-Derived Microtumors Identifies Individual Treatment Responses and Therapeutic Vulnerabilities in Ovarian Cancer"

_cancers, 2022, doi:10.3390/cancers14122895_

Round 1

Reviewer 1 Report

This paper is an interesting and innovative extension of the authors' series of papers. Ovarian cancer heterogeneity represents a key feature because of the lack of successful treatment with this tumor. For the benefit of the reader, however, a number of points need clarifying and certain statements require further justification.  These are given below;

1)In the successful cases, please describe the heterogeneity. 2) please comments why the two cases not successful in making PDM. 3) please comments why the three cases (#17, #18 and #23) in Fig.1 and the four cases (#8, #24, #25, and #26)  in supplements were chosen. 4) Please show the relationships between the successful PDM cases and TIL, leading to the co-culture cases. 5) Please show the data of TIL, resulting in co-culture cases.

Reviewer 2 Report

The authors developed patient-derived microtumors for personalized model. They also demonstrate the personalized targeted pathway by protein analysis and verify targeted drug in PDMs for therapeutic approach. Personalized model is very important for prediction of chemotherapy and immunotherapy. The authors utilize a lot of patient sample and demonstrated deep analysis especially the RPPA protein profiling and propose that PDM is one of the attractive personalized models.

Major Comments:
-The authors demonstrated immunohistochemistry of OvCa PDMs and corresponding primary tumor tissue Fig. 2 and Fig. S1. In my point of view, OvCa PDMs don’t resemble histopathological features of the corresponding primary tumor tissues. For example,MSNL on OvCa#18 and 23, Collagen1 on OvCa#24 and #25, FAPαon OvCa#24, 25 and 26. Is there any similarity index? (To clarify the similarity between OvCa PDMs and corresponding primary tumor tissues, the author should validated the degree of similarity, e.g. scoring or quantification of the staining.) Otherwise, it is hard to conclude that PDMs along with the personalized model systems are representing patient tumor.

-Fig 3 showed RPPA protein profiling of OvCa PDM. The author should add RPPA protein profiling of corresponding primary tumor tissues. If protein profiling of OvCa PDMs and corresponding primary tumor tissues are similar, PDM is acceptable for preclinical platform.
- The authors should clearly explain why PDM have advantages over the PDO. If possible, please show the comparison data of PDOs and PDMs of OvCa.

Minor comments
-Paragraph of 2.3. RPPA and protein data analysis in Materials and methods, the author repeats same sentence (Detailed methods of sample preparation and RPPA processing are provided in SI Materials). Please correct.

Reviewer 3 Report

An interesting manuscript that could be the next step in the idivualization of cancer treatment. The method of determining the sensitivity of ovarian tumors to various therapies seems quite simple and probably easy to use in practice. After completing the research on a larger group of patients, as suggested by the authors, the presented method may become an important element of the decision regarding the choice of ovarian cancer therapy.

Round 2

Reviewer 2 Report

The authors developed patient-derived microtumors for personalized model. They also demonstrate the personalized targeted pathway by protein analysis and verify targeted drug in PDMs for therapeutic approach. Personalized model is very important for prediction of chemotherapy and immunotherapy. The authors utilize a lot of patient sample and demonstrated deep analysis especially the RPPA protein profiling and propose that PDM is one of the attractive personalized models.

Major Comments:

The authors demonstrated immunohistochemistry of OvCa PDMs and corresponding primary tumor tissue Fig. 2 and Fig. S1. In my point of view, OvCa PDMs don’t resemble histopathological features of the corresponding primary tumor tissues. For example,MSNL on OvCa#18 and 23, Collagen1 on OvCa#24 and #25, FAPαon OvCa#24, 25 and 26. Is there any similarity index? (To clarify the similarity between OvCa PDMs and corresponding primary tumor tissues, the author should validated the degree of similarity, e.g. scoring or quantification of the staining.) Otherwise, it is hard to conclude that PDMs along with the personalized model systems are representing patient tumor.

 We thank the reviewer for the important advice. We have added results from immunohistochemical stainings of additional PDM models to the revised manuscript for confirmation (Figure S1).

 Additional data (Fig S1) showed only immunohistochemical staining of PDM models. Author should compare immunohistochemical staining of PDM and PTT. This additional data from PDM alone cannot support similarity of PDM and PTT.

 Furthermore, we tuned down the wording and made the description of heterogeneity of the models studied more explicit in our revised manuscript (revised paragraph 3.3). Indeed, differences in expression between microtumours and corresponding tumor tissue can be observed for some markers in studied PDM models. We have described and discussed this in more detail in the revised manuscript.

The aim of our work is the qualitative histological comparison of PDM with the corresponding primary tumor tissue (PTT). The histological comparability of PDM and PTT is confirmed by a certified pathologist based on H&E stainings (section 3.2 of our manuscript). Furthermore, for the markers p53 and WT1, which are the only markers curr

ently used for the histopathological assessment of ovarian cancer tissue, expression is comparable between PDM and PTT for the majority of cases examined in our study.

For the scoring of PDM and PTT staining, proposed by the reviewer, it is necessary to use the majority of the isolated microtumours for respective IHC staining in order to be able to make a quantitative statement. Since the amount of tumor tissue available to us is limited and restricts the amount of microtumors that can be isolated from each sample this would mean that significantly fewer microtumours of the individual models would be available for further analyses by protein profiling, compound testing and co-cultures. As these analyses represent an important part of our work, we have focused our study on qualitative investigations with regard to histology.

The author explained more details of immunohistochemistry data. However, a lot of markers are different between PDM and PTT, so the similarity between PDM and PTT that the authors have claimed are not agreeable. If author want to show that PDM can mimic the heterogeneity of tumor tissue by immunohistochemical staining, the author should show multiple area of each tissue of both PDM and PTT.

 Fig 3 showed RPPA protein profiling of OvCa PDM. The author should add RPPA protein profiling of corresponding primary tumor tissues. If protein profiling of OvCa PDMs and corresponding primary tumor tissues are similar, PDM is acceptable for preclinical platform.

 For the comparative analyses of microtumours and corresponding primary tumour tissue, 6-8 slides each with sections (4-6µm thickness) of respective primary tumor tissue were available to us. For RPPA-based protein profiling of the corresponding primary tumour tissue, more tissue material (thicker sections) is needed, which is not available to us. Therefore, we could not perform these experiments for the revised manuscript.

 I agree that the amount of primary tissues are limited and usually not enough for RPPA analysis. However, RPPA-based protein profiling of patient tissue is critical for this manuscript.  The author should show RPPA-based protein profiling of patient tissue, even from one primary tissue is acceptable.

 The authors should clearly explain why PDM have advantages over the PDO. If possible, please show the comparison data of PDOs and PDMs of OvCa.

We are grateful for this advice and have made the advantages of PDM over PDO clearer in the discussion of our revised manuscript. As the amount of fresh tumor tissue obtained for PDM isolation is limited, it is not possible for us to produce PDM and PDO cultures in parallel from the same sample.

Thank you very much for adding the discussion about the advantages of PDM over PDO. I clearly understand and accept advantages of PDM over PDO.

Minor comments

Paragraph of 2.3. RPPA and protein data analysis in Materials and methods, the author repeats same sentence (Detailed methods of sample preparation and RPPA processing are provided in SI Materials). Please correct.

We addressed this point in the revised version of the manuscript.

 Thank you very much for correction.

Author Response

Please see the attachment. Our comments are marked in "grey". 
